# Exploring the Multidimensional Participation of Adults Living in the Community in the Chronic Phase following Acquired Brain Injury

**DOI:** 10.3390/ijerph191811408

**Published:** 2022-09-10

**Authors:** Aviva Beit Yosef, Nirit Refaeli, Jeremy M. Jacobs, Jeffrey Shames, Yafit Gilboa

**Affiliations:** 1School of Occupational Therapy, Faculty of Medicine, Hebrew University of Jerusalem, Jerusalem 9124001, Israel; 2School of Medicine, Faculty of Medicine, Hebrew University of Jerusalem, Jerusalem 9112102, Israel; 3Department of Geriatrics and Geriatric Rehabilitation, Hadassah Medical Center, Jerusalem 9124001, Israel; 4Medical and Health Professions Division, Maccabi Health Services, Tel Aviv 6812509, Israel

**Keywords:** stroke, traumatic brain injury, ICF, community reintegration, disability, occupational therapy, occupational gaps, rehabilitation, client-centered, environmental factors

## Abstract

This exploratory study aimed to examine multiple aspects of the participation of adults in the chronic phase following acquired brain injury (ABI), considering different disability levels. Our study included 25 adults ≥6 months after ABI (predominantly stroke), living at home, without severe cognitive decline. Primary measures included the Canadian Occupational Performance Measure (subjective participation) and the Mayo-Portland Adaptability Inventory-4 Participation Index (objective participation). The results indicated subjective participation problems in all of the International Classification of Functioning, Disability and Health participation domains. In addition, objective participation was reported as most limited in the areas of leisure and recreational activities, residence, and employment. Both subjective and objective participation profiles varied according to the disability level except for the social and leisure areas, which were found to be similar across all subgroups. However, only partial compatibility was found between the subjective and objective participation aspects. To conclude, our findings indicated that chronic ABI survivors report a variety of subjective and objective participation concerns that varied according to their disability levels. Moreover, the incongruity between the participation aspects suggests that the level of limitation may not necessarily correspond to the importance of a particular participation area. This highlights the need for comprehensive assessments to determine unique individual participation profiles in order to facilitate client-centered interventions supporting the rehabilitation of community-dwelling ABI survivors.

## 1. Introduction

The International Classification of Functioning, Disability and Health (ICF) defines participation broadly as involvement in life situations. This includes roles and activities in all aspects of daily life, from basic to more complex activities in the home and in the community. According to the ICF, activities can be classified into nine broad participation domains, including: (1) learning and applying knowledge; (2) general tasks and demands; (3) communication; (4) mobility; (5) self-care; (6) domestic life; (7) interpersonal interactions and relationships; (8) major life areas; and (9) community, social, and civic life [1]. An important innovation of the ICF was its inclusion of contextual factors that influence participation, including environmental and personal factors [1,2,3,4]. The ICF can serve as a basis for common terminology for health documentation [1,5,6].

The concept of participation is multidimensional, and the literature on the subject generally describes two dimensions of participation: objective and subjective (see Figure 1). The objective dimension includes practical and observable aspects, such as the frequency of performance and limitation levels. The subjective dimension involves the person’s feelings and self-perceptions regarding participation, including aspects such as satisfaction and importance of activities, which are affected by the person’s interests, roles, preferences, and past experiences [7,8,9,10,11,12].

Participation is a central concept in health care and is widely regarded as the ultimate goal of rehabilitation following acquired brain injury (ABI) [13,14]. ABI generally refers to brain injuries sustained after birth, commonly caused by stroke or traumatic brain injury (TBI) [15,16]. ABI is a major health issue and a leading cause of long-term functional disability and participation restrictions in daily life [17,18,19,20,21]. In light of the outbreak of COVID-19, which disrupted individuals’ daily activities and routines [22,23], it appears that this issue has been exacerbated. A recent study found that people with ABI reported a reduction in activity participation during the pandemic compared with their pre-pandemic engagement and compared with healthy adults [24]. Several studies have found that participation is a powerful predictor of quality of life and life satisfaction following ABI [20,24,25,26,27]. Given the importance of participation as a rehabilitation outcome, and in accordance with the spirit of the ICF, there is an increasing interest in understanding the participation of people with ABI. There has been extensive research on the factors influencing participation after an ABI. Some of the factors that have emerged as contributing to community participation outcomes across several studies include the level of disability [6,27,28,29,30,31] and environmental characteristics [31,32,33,34].

A majority of the studies that examined participation after ABI focused mostly on the objective aspects such as frequency and limitation level. The typical participation profile that emerges from these studies indicates long-standing participation restrictions compared with pre-injury participation and compared with participation among healthy adults. Low performance frequency and high limitation levels are described in various domains including self-care activities (less reported after mild ABI) and mainly in the areas of domestic life, employment, mobility, leisure, and social activities [7,9,10,11,20,24,27,29,30,32,35,36,37,38,39,40,41,42,43,44,45,46]. The current literature that used the ICF terminology to describe participation focused mostly on the objective aspects [6,45,46,47,48,49,50,51], while the subjective aspects are notably absent.

Studies that have examined the subjective dimension of participation are fewer, and they have mostly focused on the satisfaction aspect. Findings indicate that satisfaction ratings in the chronic stage post-ABI are generally low but vary between domains. Generally, a low level of satisfaction is reported in the areas of self-care, domestic life, employment, leisure, and outdoor activities; less dissatisfaction is reported in the areas of family life and social relationships [7,9,30,35]. There are few studies that considered additional subjective aspects such as meaningfulness, importance, and desire to change. Some differences were found in these dimensions between younger and older adults following ABI [11,52]; however, no significant differences were found between different disability levels [10]. Eriksson et al. [41] found that the number of reported occupational gaps correlated moderately with self-rated participation. Johnston et al. [53] examined the relationships between two subjective aspects of participation and found a robust correlation between dissatisfaction with various activities and the desire to change them but not one so strong that these two aspects can be considered identical. A subjective aspect of participation that received less attention is the issue of prioritized activities from the perspective of the individual, in other words, not just what is difficult or what one would like to do more often but what are the most important activities. Kersey et al. [32] aimed to understand the prioritized activities identified by people following TBI. They found that participants prioritized activities related to community participation (paid or unpaid work, socialization, recreational activities, and learning) rather than activities related to personal care or mobility. Additionally, the participants expressed a desire for change in all domains, particularly in social participation. This issue is paramount since ABI survivors who participate in activities that are important to them experience pleasure, satisfaction, and a sense of belonging, which are associated with improved health and well-being [10,54,55]. Therefore, it is necessary to gain a better understanding of clients’ valued and desired activities for effective rehabilitation planning and goal setting [11].

Several studies included both objective and subjective measures of participation and explored the relationships between these aspects. Some of these studies found weak to moderate correlations between objective aspects of participation such as frequency and limitation levels and the subjective aspect of satisfaction with participation in the chronic stage of ABI recovery [7,9,35,39,41,53]. However, Eriksson et al. [56] found a more robust correlation between participation limitation and satisfaction among adults with mild stroke living in the community. Similarly, Toglia et al. [11] found a moderate positive correlation between the frequency of participation and the engagement in meaningful activities among adults post stroke. In addition, Kersey et al. [32] recently examined patterns of the community participation of adults with TBI. They found that perceived activity difficulty and frequency were both impaired across all participation domains and that participants consistently expressed a desire for change across these domains, particularly in the domains of social participation and productivity. However, Cheraghifard et al. revealed that objective and subjective participation differed in their ability to differentiate between disability levels among chronic stroke survivors; whereas frequency of participation differentiated between different levels of disability, the degree of meaningfulness of activities did not [10].

In summary, despite the consensus regarding the use of the ICF, there is still inconsistency in the terminology used to describe participation and measure it. Participation as a construct is ambiguous, and measurement methods are inconsistent [8,32]. Moreover, the diverse findings regarding the objective and subjective dimensions of participation following ABI underscore the need for and importance of understanding and exploring this concept in a multidimensional way for varying disability levels. This may contribute to the conceptualization of participation and its application in designing and evaluating interventions for community-dwelling ABI survivors in the chronic stage [11,13,57]. Therefore, the aim of our exploratory study was to examine multiple aspects of participation within a sample of community-residing adults in the chronic stage post-ABI considering different disability levels. Specific aims were to: (a) describe the subjective participation (importance) using the ICF terminology, (b) describe the objective participation (limitation level); (c) explore the compatibility of the objective and subjective aspects of participation; and (d) explore the relationships between the different aspects of participation and the perceived environmental accessibility.

## 2. Materials and Methods

### 2.1. Study Design and Participants

The current study was an exploratory cross-sectional study with descriptive and analytical methods. This study included the baseline data collected as part of a research project devoted to the development and assessment of a home-based telerehabilitation program for community-dwelling adults in the chronic stage following acquired brain injury (ABI) [58,59].

People who met the following inclusion criteria were included in the study: (1) ≥6 months after ABI; (2) age ≥ 18 years; (3) sufficient level of Hebrew or English to participate in this study; (4) modified Rankin scale (mRS) scores of 2–4 reflecting slight to moderately severe disability [60]. Participants were excluded with the following criteria: (1) moderate or severe aphasia; (2) dementia diagnosis or a score of <21 on the Mini Mental Status Examination (MMSE) [61] or <19 on the Montreal Cognitive Assessment (MOCA) [62,63]; or (3) an acute illness which significantly impacts the ability to participate in the study.

### 2.2. Procedure

We recruited participants from three day-rehabilitation centers in Israel, following approval by the research ethics committees of the Hadassah-Hebrew University Medical Center, Jerusalem, and Maccabi Healthcare Services, Bat-Yam, Israel (ethical committee registration numbers: 0689-15-HMO and 192016, respectively). The study was conducted according to the Declaration of Helsinki, with informed consent provided by all participants. The current study was performed as part of the baseline assessment in our intervention study [59]. Assessment was conducted in the participants’ homes by two licensed occupational therapists with more than five years of experience in geriatric and neurological rehabilitation and was performed in two sessions (approximately 1.5 h each session). After the first evaluation session, two participants withdrew from the study because of personal or health issues not related to the study. For these participants, we used the partial data collected.

### 2.3. Measures

#### 2.3.1. Sociodemographic and Clinical Characteristics

All outcome measures were administered in validated Hebrew versions. The sociodemographic characteristics of the participants were documented using a background questionnaire developed for the study. Clinical characteristics were documented as well, using the background questionnaire and through reviews of medical records. As part of our background questionnaire, we investigated community mobility using the frequency of leaving one’s home. We asked how often the participants left the house (how many days a week). Answers were categorized as daily or nearly daily (6–7 days a week), often (2–5 days a week), or rarely (≤1 time a week) [64,65].

We also documented the disability levels of the participants based on the mRS ratings. The mRS assesses the level of disability of people with neurological diseases on a 7-level scale: 0—no symptoms; 1—no significant disability; 2—slight disability; 3—moderate disability; 4—moderately severe disability; 5—severe disability; 6—dead. In this study, we specifically used the mRS-9Q, which measures the mRS scores in neurological patients using a nine-question “yes/no” survey. Calculation and error checking of the mRS score were performed using a web-based tool [60].

Additional measures were used to describe other clinical characteristics of the sample. We measured depressive symptoms with either the Personal Health Questionnaire (PHQ-9) [66] or the Geriatric Depression Scale (GDS) [67] and noted the severity level of the symptoms (e.g., mild) to maintain uniform reporting (the PHQ was added so that younger participants could also be included). Executive function in daily life was measured using the Dysexecutive Questionnaire (DEX) [68]. On the DEX, there are 20 items, each rated from 0 to 4, according to how frequently the problems manifest in everyday life. A higher score indicates more executive function problems in daily life [68]. The DEX was found to possess sufficient concurrent [69] and ecological validity and good internal consistency (Cronbach’s α = 0.89) [70]. In addition, the DEX significantly distinguished individuals with brain injury of various etiologies from healthy controls [69,70,71].

#### 2.3.2. Participation Measures

Subjective participation was measured with the Canadian Occupational Performance Measure (COPM) [72]. The COPM is a semi-structured interview that facilitates client-centered goal setting. It measures the client’s perceived performance and satisfaction levels for prioritized occupational performance problems identified during the interview. The five most important occupational performance problems are rated using a 10-point performance scale (1—not able to do it, 10—able to do extremely well) and satisfaction with performance scale (1—not satisfied at all, 10—extremely satisfied). The COPM is widely used among adults after ABI and has been shown to be a reliable and valid outcome measure [72,73,74,75].

Objective participation was measured using the Mayo-Portland Adaptability Inventory-4th Edition-Participation Index (MPAI-4-P). The MPAI-4 is widely used for assessing the recovery progress of people after an ABI and consists of three subscales: (a) ability, (b) adaptation, and (c) participation. The participation index, which was used in this study, includes eight items that represent different participation areas. These include initiation, social contact, leisure and recreation, self-care, residence, employment, transportation, and managing money and finances. The items are rated on a scale of 0–4, with higher scores indicating lower participation [76,77]. Scores are converted into T-scores representing different participation limitation levels: scores beneath 30 indicate relatively high participation; scores between 30 and 40 indicate mild limitations; scores between 40 and 50 indicate mild to moderate limitations; scores above 60 indicate severe limitations [77]. It has been well established that the MPAI-4 provides satisfactory internal consistency (Cronbach’s *α* = 0.85–0.90) [78] and high construct, concurrent, and predictive validity for the full questionnaire and its subscales [76,79]. Furthermore, the MPAI-4 shows sensitivity to clinical change after rehabilitation [80,81].

#### 2.3.3. Environmental Factor: Perceived Accessibility

A brief measure of the perceived accessibility of the participants’ environment was developed for the study with the intention of capturing the subjective component of accessibility. Two items were used: one to measure the accessibility of the home environment and the other to measure the accessibility of the community environment. The participants were provided with a brief explanation and examples illustrating each concept and were asked to rate each item on a 10-point rating scale (1—not accessible, 10—fully accessible).

### 2.4. Statistical Analysis

Descriptive statistics were used to describe the characteristics of the sample, as well as various aspects of participation, based on the data from the objective and subjective participation outcome measures. Nonparametric statistics were used due to the small sample size. A Spearman correlation test was used to investigate the relationships between the measures. The independent-samples Kruskal–Wallis Test was used to compare the sociodemographic and clinical characteristics of the three mRS level subgroups for continuous data and the Fisher’s exact test for categorical data. Analysis was conducted with SPSS Version 27.0 (IBM Corporation, Armonk, NY, USA). Significance was set at *p* < 0.05 for all analyses. The statistical analyses were not corrected for multiple testing due to the exploratory nature of the study. For the analysis of the occupational performance problems identified using the COPM according to the ICF domains, two occupational therapists (authors ABY and NR) independently classified each occupational performance problem to the appropriate participation domain according to ICF’s accepted linkage rules [82]. The classifications were compared and checked for matching, and discrepancies were resolved through joint discussion until agreement was reached.

## 3. Results

### 3.1. Sample Characteristics

Data from 25 participants were analyzed to explore the participation of adults in the chronic phase living in the community after ABI. Table 1 presents the sociodemographic and clinical characteristics of the study sample. The average age of the participants was 60 (±10.70) years old; however, there was a wide range of ages (35–79 years). There were more men (60%) than women (40%), and the education level was fairly high (*M* = 13 ± 3.55 years). The majority of participants were married (80%) and living with their partner or family (88%) in an urban area (88%). Among the participants, 6 (24%) were employed post-ABI either part-time or full-time, compared with 17 (68%) who were employed before the ABI (8 of whom were over 60). Accordingly, 11 of 17 participants (65%) ceased to work following the ABI.

As for the clinical characteristics of the sample, the most frequent type of injury was an ischemic stroke (68%), and the average time since the injury was 9.44 (±2.95) months. According to the mRS scores, three groups of similar size were formed, with the subgroup of limitation level 3 being relatively larger (40%) than the other two subgroups. Statistical analysis revealed no significant differences between the mRS subgroups in the sociodemographic characteristics except for years of education (subgroup mRS 2 had significantly more years of education than the other two subgroups). Mental status based on cutoff depression screening scores indicated that 38% of the participants reported some level of depression. Regarding the cognitive status, there were relatively low DEX scores (*M* = 15.67 ± 12.84), which could suggest that the participants did not experience executive function problems in daily life very often.

Prior to the ABI, most of the participants left the house daily or nearly daily (96%), drove (72%), and walked outside as a method of community mobility (96%). Following the ABI, all participants were able to walk independently inside their homes, either with or without a walking aid. However, 24% of the participants reported not walking outside as a means of community mobility, and only 32% reported that they drove. Moreover, less than half (44%) of the participants reported leaving the house on a daily or nearly daily basis, and 16% reported they left the house once a week or less. Finally, on a scale of 1–10, participants reported relatively high perceived accessibility of their home environment (*M* = 8.88 ± 1.36) and slightly lower perceived accessibility of their community environment (*M* = 7.38 ± 2.07).

### 3.2. Subjective Participation According to the International Classification of Functioning, Disability and Health (ICF) Terminology

The five most important occupational performance problems were identified by the participants using the COPM. Overall, a total of 121 prioritized occupational problems were identified. On a 1–10 scale, participants’ mean importance rating was 9.13 (± 0.75); their mean performance and satisfaction ratings were (*M* = 3.3 ± 1.06) and (*M* = 3.16 ± 1.23) respectively. In order to describe the subjective aspects of participation in terms of importance, we linked the prioritized occupational problems to the corresponding ICF categories (https://apps.who.int/classifications/icfbrowser/Default.aspx, accessed on 13 July 2020). Table 2 includes examples to illustrate how occupational problems were analyzed and categorized into three levels of ICF activity and participation domains.

The results showed that adults in the chronic stage after ABI face diverse occupational performance problems. Figure 2 shows the distribution of participants’ most important occupational problems according to the ICF domains. As can be seen in Figure 2(a), when examining the whole sample (*N* = 25) the ICF domain with the most prioritized occupational problems was ‘Self-care’ (*n* = 32). However, it is important to note that 13 of them (41%) were in the area of ‘Looking after one’s health’, whereas the other 19 (59%) were problems related to basic activities of daily living (BADL) including ‘Dressing’, ‘washing oneself’, ‘eating’ and ‘caring for body parts’ (as can be seen in Appendix A Table A1). The other domains in which the most prioritized occupational problems were identified were ‘domestic life’ (*n* = 23), ‘community, social and civic life’ (*n* = 22), and ‘major life areas’ (*n* = 19; mainly employment related). The domains framed by the blue line were the most prevalent. Across the various ICF domains, the mean importance ratings ranged between 8.5 and 10, reflecting the high importance the participants attached to their prioritized occupational problems.

The examination of the prioritized occupational performance problems according to the different mRS subgroups revealed that the proportions of the ICF domains shifted. As can be seen in Figure 2b, in the mRS 2 and mRS 4 subgroups, the most prevalent problems were in the ‘self-care’ domain; however, in the mRS2 subgroup 75% of the ‘self-care’ problems were in the subdomain of ‘looking after one’s health’. On the other hand, in the mRS 4 subgroup, the majority of the ‘self-care’ problems (83%) were BADL problems. The results also indicated that in the mRS 4 subgroup, there was only one occupational problem related to ‘major life areas’, whereas in the other two subgroups, there were more problems related to ‘major life areas’ such as employment (19–24%). Interestingly, the results indicated that there were similar percentages of occupational problems related to ‘interpersonal interactions and relationships’ (6–16%) and ‘community, social and civic life’ domains (18–19%) across disability levels.

### 3.3. Objective Participation

In regard to the objective aspect of participation (limitation level), the mean MPAI-4-P score was 49.4 (±10.91), indicating mild to moderate participation limitations. To gain a deeper understanding of the characteristics of participation beyond the final score, questionnaire items were analyzed in order to describe the participation of the sample according to participation areas. Figure 3 presents the mean score for each item on the MPAI-4-P. The results showed that the participants (*N* = 22; green bars) rated the lowest participation (higher scores) for ‘leisure and recreation’ (*M* = 1.9 ± 1.4), ‘residence’ (*M* = 2.1 ± 1.3), and ‘employment’ (*M* = 2.3 ± 1.5).

When divided into subgroups according to mRS disability levels, the distribution of ratings differed between subgroups, as also shown in Figure 3. The ratings of the subgroups for most items corresponded to the mRS disability levels, with the exception of the ‘social contact’ item, where the limitations were relatively low and similar between the subgroups. The results indicated that the mRS 4 subgroup reported the lowest participation for most items. It is very noticeable that this group rated many items as very limited, especially ‘managing money and finances’, ‘employment’, ‘transportation’, ‘residence’, and ‘leisure and recreation’. In general, the mRS 2 subgroup ratings were less severe, with the highest limitation scores reported in ‘employment’ and ‘leisure and recreation’ and the lowest participation limitation scores reported in ‘self-care’ and ‘managing money and finances’. The middle subgroup, mRS 3, reported participation limitations mainly in the areas of ‘employment’, ‘residence’, ‘transportation’, and ‘leisure and recreation’.

### 3.4. Exploration of the Compatibility of the Objective and Subjective Aspects of Participation

A Spearman’s test was conducted to explore the correlations between the MPAI-4-P and the COPM scores. The results indicated nonsignificant correlations between COPM performance and MPAI-4-P scores (*r_s_* = −0.17, *p* = 0.443) and between COPM satisfaction and MPAI-4-P scores (*r_s_* = −0.38, *p* = 0.087).

Additionally, we classified the occupational problems as identified by the COPM by areas of participation according to the MPAI-4-P items in order to explore whether the areas with higher limitation levels also had more prioritized occupational performance problems. This analysis included all the participants who completed both the COPM and the MPAI-4-P (*N* = 22). It should be noted that one occupational problem was not matched with any item. Figure 4 illustrates the mean rating of each MPAI-4-P item along with the percentage of prioritized occupational problems that matched with each item.

When analyzing the results of the whole sample, there was partial compatibility between the MPAI-4-P ratings and the percentage of prioritized occupational performance problems identified using the COPM. In other words, a high rating for some participation area (which reflects low participation) corresponded to a relatively high percentage of prioritized occupational problems in the same area, and vice versa. This was most evident in the three items with the highest participation limitation ratings, which also had the most prioritized occupational problems associated with them: ‘leisure and recreation’ (*M* = 1.9 ± 1.8; 19.6%), ‘residence’ (*M* = 2.1 ± 1.3; 26.2%), and ‘employment’ (*M* = 2.3 ± 1.5; 17.8%). On the other hand, despite a relatively high limitation rating for ‘transportation’ (*M* = 1.7 ± 1.4) and ‘managing money and finances’ (*M* = 1.6 ± 1.6), only a small percentage of prioritized occupational problems were identified in these domains (3.7% and 5.6%, respectively). In addition, there were no occupational problems associated with the ‘initiation’ item, even though this item was reported to have some limitations (*M* = 1.2 ± 1.1).

Similarly, when we examined the results according to the mRS disability levels, we found some incompatibilities as seen in Figure 5. For example, the mRS 2 subgroup rated the area of ‘residence’ as having low participation limitations (*M* = 0.5 ± 0.84); however, it is apparent that this area had the highest percentage (25.0%) of prioritized occupational performance problems. Another example can be found in the mRS 4 subgroup, where the highest percentage of prioritized occupational problems was attributable to ‘self-care’ (43.4%), which was not the area with the highest participation level limitation ratings (*M* = 2.67 ± 1.21). In this subgroup, four other areas had more severe participation limitations, ‘managing money and finances’ (*M* = 3.00 ± 1.55), ‘employment’ (*M* = 3.67 ± 0.82), ‘transportation’ (*M* = 2.83 ± 1.33), and ‘residence’ (*M* = 3.33 ± 0.82); however, these areas had fewer prioritized occupational problems (6.7%, 10.0%, 3.3%, 20%, respectively). The mRS 3 subgroup showed more compatibility, as the three areas with the most limited participation levels matched the areas with the most prioritized occupational problems: ‘leisure and recreation’ (*M* = 1.7 ± 1.25; 24.5%), ‘residence’ (*M* = 2.20 ± 0.63; 30.6%), and ‘employment’ (*M* = 1.80 ± 1.55; 24.5%).

### 3.5. Exploration of the Relationship between the Perceived Environmental Accessibility and the Different Aspects of Participation

A Spearman’s test was conducted to explore the correlations between the perceived accessibility ratings and the scores on the MPAI-4-P (objective participation; limitation level) and COPM (subjective participation; performance and satisfaction of prioritized occupational performance problems). The results indicated significant negative correlations between the perceived accessibility ratings of both the home and community environments and the MPAI-4-P scores. The negative correlations indicated that participants who perceived the home and community as less accessible reported more participation limitations (higher scores on the MPAI-4-P scale). However, no significant correlations were found between the perceived accessibility ratings and the COPM scores (see Table 3).

## 4. Discussion

Our exploratory study aimed to examine objective and subjective dimensions of participation among adults in the chronic phase following ABI considering different disability levels. Our examination of subjective participation by highlighting the prioritized occupational performance problems revealed that the participants reported a wide range of prioritized occupational issues that were distributed differently according to disability levels. Similarly, participants’ reports regarding the objective participation dimension showed limited participation in various areas distributed differently according to the subgroups. A finding that stood out was the similarity of the reports regarding the social domain in both the objective and subjective dimensions across disability levels. However, we found that despite the wide range of participation domains affected in both the objective and subjective participation dimensions, there was only partial compatibility between them in each participation domain.

Similar to the results of the present study, previous studies investigating the participation of adults in the chronic phase following ABI have consistently found a negative effect on objective and subjective participation in a variety of domains [7,9,10,11,20,24,27,29,30,32,35,36,37,38,39,40,41,42,43,44,45,46,52]. Moreover, we found that despite focusing only on the prioritized occupational issues, it was still possible to link them to all nine participation domains of the ICF. This result is in line with several studies describing the therapeutic goals or needs identified by chronic ABI survivors, indicating that the prioritized occupational issues are from all participation domains [75,83,84,85,86].

In our study, the ICF domain with the most prioritized occupational problems was ‘self-care’. Interestingly, the most prominent category reported in this domain was ‘looking after one’s health’. In contrast to our results, Kersey et al. [32] found that participants prioritized leisure, employment, and socialization activities over personal care and mobility activities. This difference between the findings is possibly due to the different definitions of self-care used by Kersey et al., who included only BADL (e.g., dressing, bathing, eating). In addition, their sample included younger participants (*M* = 42.7 ± 17.3 years) with mild physical and mobility symptoms. Notably, the area of health management does not always appear in common participation measures such as the stroke impact scale participation subscale [87], the Utrecht Scale for Evaluation of Rehabilitation-Participation [88], or the Community Participation Indicators [89]. Thus, there is a possibility that literature regarding the participation repertoire of this population has not adequately addressed the important issue of health management, and it should be highlighted and investigated further. This finding underscores the importance of promoting self-management programs among people after ABI [90,91,92,93].

A novel aspect of our research is its examination of the participation profiles according to different disability levels. A closer look at the participation patterns of the subgroups revealed that there were some differences in the distribution of the prioritized occupational problems among the domains. This stood out especially in the ‘self-care’ domain, which was prominent in the group who had a more severe disability level, and the ‘major life areas’ domain (primarily employment issues), which was more prominent in the groups with lower disability levels. Despite these differences, the domains of ‘interpersonal interactions and relationships’ and ‘community, social and civic life’ had similar and relatively large proportions of prioritized occupational problems across the three disability levels. This may indicate that these domains are significant regardless of the disability level and is in line with other studies that have found that these domains remain problematic and important even at the chronic stage following ABI [11,32,41,75,84,86]. This result adds to the previous literature indicating that individuals with ABI are especially vulnerable to social isolation and loneliness [94], and this direction should be further studied in future research.

Regarding the objective aspect of participation, unsurprisingly, our results showed that in most participation areas, participants with a greater level of disability reported more severe participation restrictions. These results are in line with previous studies that have reported an association between disability level and participation restrictions [6,27,28,29,30,31]. It stood out, however, that the three groups reported similar and relatively high participation levels in the social area. This is in accordance with other studies that found low levels of restrictions in the social domain [7,9]. In contrast, Silva et al. [6] found that the social area was the most severely affected after stroke. This could be explained by the differences in marital status between the samples. The majority of our sample, 80%, was married, compared with Silva et al.’s sample, nearly half of whom were unmarried, and their results showed that widowed individuals had significantly lower participation levels than married individuals [6].

We found partial compatibility between the objective and subjective participation dimensions in our study. If one looks at the entire sample, there seems to be a match between the two participation aspects, even if it is imperfect; however, when considering each disability level subgroup separately, the gap becomes more apparent. In addition, the lack of correlation found between the objective and subjective aspects of participation points to a similar conclusion, that the level of limitations people report in various areas does not fully match areas that are most important to them. These results support the contention that objective participation is not necessarily indicative of subjective participation or vice versa. These findings align with results from qualitative studies that explored the concept of meaningful participation and the fundamental impact it has on the lives of ABI survivors [2,14]. An individual can experience full and meaningful subjective participation even if their objective participation is incomplete or lacking. Having one important and significant role can give a sense of full participation even without returning to all the participation areas [2]. Thus, this conclusion underscores the need for clinicians to include different measures to reflect the multidimensional aspects of participation; by doing so, a more comprehensive picture of people’s participation will emerge [7,9,11,52]. Furthermore, this message also echoes the principle of client-centered care that focuses on the personal, significant, and unique needs of an individual [57,95,96] and reflects the current paradigm shift in health care toward more holistic and patient-centered care [97].

Finally, we found an association between home and community accessibility and objective participation, indicating that greater environmental accessibility is associated with fewer participation limitations. This finding is in line with results of previous studies that showed that environmental characteristics contributed to community participation [31,32,34]. Moreover, this corresponds to the ICF model portraying the interaction between the environmental context and participation [1]. In contrast, we did not find a correlation between home and community accessibility and subjective participation. This reaffirms the distinction between the two dimensions of participation. Additionally, as other authors have noted, the COPM scoring method does not lend itself well to generating large variance in the data, which can result in the lack of correlations, as found in our study [86,98].

We must acknowledge some limitations of this exploratory study that require caution when interpreting its results. First, a major limitation of the study was the relatively small sample, which was smaller than we planned since the recruitment was significantly harmed due to the COVID-19 outbreak. This affected the size of the subgroups as well, and furthermore limited the possibility of multiple comparisons. In addition, despite the heterogeneity of the sample in terms of disability level and etiology, it is likely that the most severe cases of ABI are underrepresented, limiting the generalizability of our findings. Furthermore, the subgroup with the lowest level of disability had a significantly higher education level. Since education has been found to be a significant factor in participation following ABI [99], this discrepancy may be an alternative explanation for the different participation profiles among the subgroups. Finally, a unique aspect of this study was its exploration of participation from both subjective and objective perspectives. However, the measures’ characteristics made comparison difficult. The use of participation measures based on the ICF participation domains may contribute to this issue in future studies. While the study has limitations, the results point to a significant issue in the community rehabilitation of people with ABI that requires further exploration, and research on a larger scale is warranted.

## 5. Conclusions

Our study adds to the large body of evidence that describes the long-term effects of ABI on the participation of adults living in the community, demonstrating the need for rehabilitation services at this stage. In addition, a novel aspect of this study is in its contribution to the evolving literature that examines participation in a multidimensional way. Our results indicate that the participation profiles varied according to the level of disability in both objective and subjective aspects of participation and that the fit was only partial between these two aspects. This emphasizes the notion that participation is not just about quantity or limitation level; rather, we must understand each person’s unique participation profile, which reflects the domains and roles most dear to them. For example, the COPM is suitable for this purpose. This point of view can facilitate holistic and client-centered interventions contributing to the recovery of ABI survivors in their homes and communities.

## Figures and Tables

**Figure 1 ijerph-19-11408-f001:**
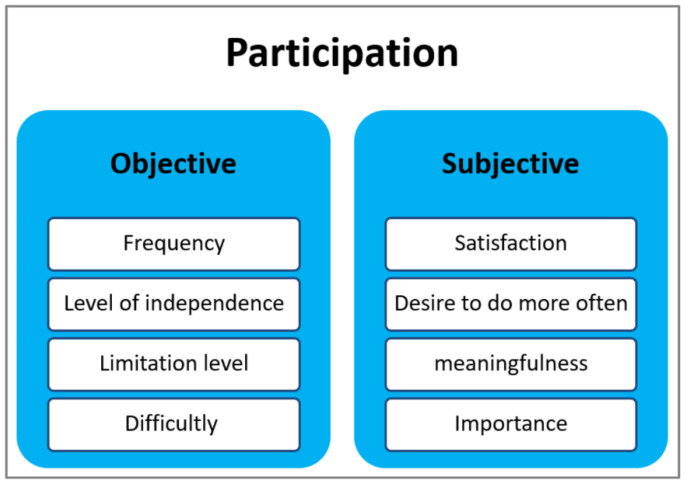
Objective and subjective dimensions of participation.

**Figure 2 ijerph-19-11408-f002:**
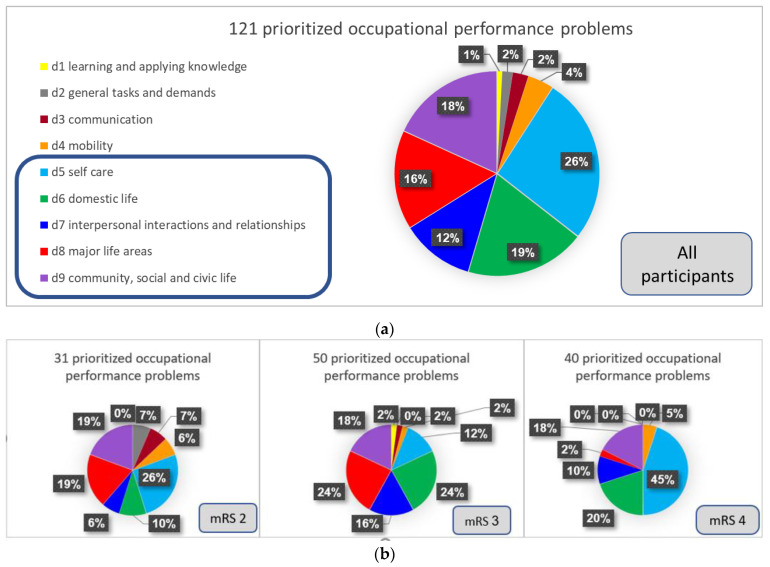
The distribution of participants’ most important occupational performance problems according to International Classification of Functioning, Disability and Health (ICF) domains: (**a**) the whole sample (*N* = 25); (**b**) subgroups according to modified Rankin scale (mRS) scores: mRS 2 (*n* = 7); mRS 3 (*n* = 10); mRS 4 (*n* = 8).

**Figure 3 ijerph-19-11408-f003:**
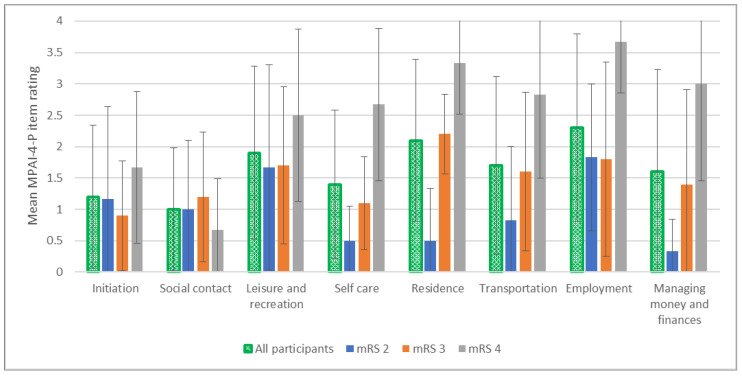
Mean Mayo-Portland Adaptability Inventory-4th edition-Participation Index (MPAI-4-P) scores per item (All participants, *n* = 22; mRS 2, *n* = 6; mRS 3, *n* = 10; mRS 4, *n* = 6); mRS, modified Rankin scale.

**Figure 4 ijerph-19-11408-f004:**
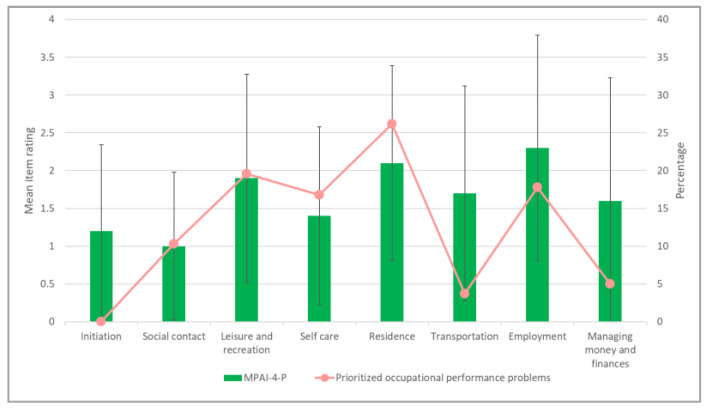
Compatibility of objective and subjective participation aspects. The figure illustrates the mean Mayo-Portland Adaptability Inventory-4th edition-Participation Index (MPAI-4-P) item scores and the percentage of prioritized occupational performance problems matched to each item (*N* = 22; 107 prioritized occupational performance problems).

**Figure 5 ijerph-19-11408-f005:**
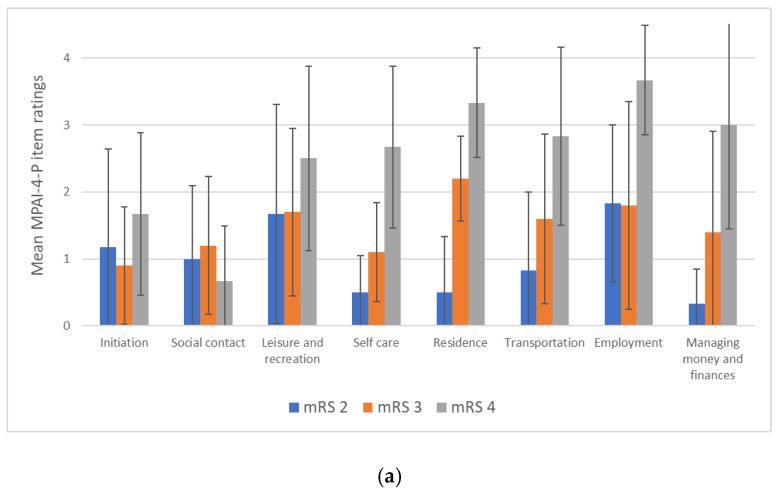
Compatibility of objective and subjective participation aspects according to modified Rankin scale (mRS) subgroups (mRS 2, *n* = 6; mRS 3, *n* = 10; mRS 4, *n* = 6): (**a**) mean Mayo-Portland Adaptability Inventory-4th edition-Participation Index (MPAI-4-P) item scores; (**b**) percentage of prioritized occupational performance problems matched to each item.

**Table 1 ijerph-19-11408-t001:** Sociodemographic and clinical characteristics of the sample (*N* = 25).

Characteristic	Mean ± *SD* (Range) or *n* (%)
Age (years)	60.84 ± 10.70 (35-79)
Sex Female Male	10 (40%) 15 (60%)
Years of education	13 ± 3.55 (2–19)
Marital status Married Separated/Divorced/Widowed	20 (80%) 5 (20%)
Living area Urban Rural	22 (88%) 3 (12%)
Employment status post-ABI Full/Part-time work Retirement/Volunteering Unemployment	6 (24%) 13 (52%) 6 (24%)
ABI type Ischemic stroke Hemorrhagic stroke Traumatic brain injury	17 (68%) 6 (24%) 2 (8%)
ABI side Right Left Bilateral	14 (56%) 10 (40%) 1 (4%)
Time since ABI (months)	9.44 ± 2.95 (6–18)
mRS scores Score 2—Slight disability Score 3—Moderate disability Score 4—Moderately severe disability	3.04 ± 0.79 (2–4) 7 (28%) 10 (40%) 8 (32%)
Depression * No Yes	13 (62%) 8 (38%)
DEX *	15.67 ± 12.84 (0–46)
Walking aid No Yes	12 (48%) 13 (52%)
Mobility outside by foot post-ABI Yes No	19 (76%) 6 (24%)
Mobility by car post-ABI Driver Passenger No car	8 (32%) 14 (56%) 3 (12%)
Frequency of leaving the house Daily or nearly daily (6–7 times a week) Often (2–5 times a week) Rarely (≤1 time a week)	11 (44%) 10 (40%) 4 (16%)
Perceived accessibility of the environment Home environment Community environment	8.88 ± 1.36 (6–10) 7.38 ± 2.07 (4–10)

*n*, number; *SD*, standard deviation; ABI, acquired brain injury; mRS, the modified Rankin Scale; DEX, Dysexecutive Questionnaire. * Cognitive and depression status (*N* = 21; missing data due to partial assessment). Depression status was based on scores on the Geriatric Depression Scale (*n* = 10) or the Patient Health Questionnaire (PHQ-9) (*n* = 11).

**Table 2 ijerph-19-11408-t002:** Examples of the classification process from occupational performance problems to the International Classification of Functioning, Disability and Health (ICF) domains.

COPM-Occupational Problem	ICF 1st-Level Domain	ICF 2nd-Level Domain	ICF 3rd-Level Domain
Write more clearly	d1. Learning and applying knowledge	d170. Writing	d1708. Writing clearly
Get organized in the morning at a faster pace	d2. General tasks and demands	d230. Carrying out daily routine	d2303. Managing one’s own activity level
Maintain my concentration during a conversation	d3. Communication	d350. Conversation	d3501. Sustaining a conversation
Getting back to riding my bicycle every day	d4. Mobility	d475. Driving	d4750. Driving human-powered transportation
Start going to a weekly exercise class again	d5. Self-care	d570. Looking after one’s health	d5701. Managing diet and fitness
Put on my pants independently	d5. Self-care	d540. Dressing	d5400. Putting on clothes
Being able to care for my dog more independently at home	d6. Domestic life	d650. Caring for household objects	d6506. Taking care of animals
Participate in weekly leisure activities with my children	d7. Interpersonal interactions and relationships	d760. Family relationships	d7600. Parent-child relationships
Return to work	d8. Major life areas	d845. Acquiring, keeping, and terminating a job	d8451. Maintaining a job
Learn how to track my bank account online	d8. Major life areas	d870. Economic self-sufficiency	d8700. Personal economic resources
Get back to reading books every day	d9. Community, social and civic life	d920. Recreation and leisure	d9202. Arts and culture
Get back to visiting the community center a few times a week	d9. Community, social and civic life	d910. Community life	d9100. Informal associations

**Table 3 ijerph-19-11408-t003:** Correlations between the mean perceived accessibility ratings and the mean participation scores.

		Objective Participation (*N* = 22)		Subjective Participation (*N* = 25)			
		MPAI-4-P		COPM-Performance		COPM-Satisfaction	
		*r_s_*	*p*	*r_s_*	*p*	*r_s_*	*p*
Perceived environmental accessibility (*N* = 22)	Home	−0.474 *	0.026	−0.065	0.757	−0.146	0.485
	Community	−0.464 *	0.030	0.056	0.790	0.166	0.429

* Significant correlation, *p* < 0.05; MPAI-4-P, Mayo-Portland Adaptability Inventory-4th Edition-Participation Index; COPM, the Canadian Occupational Performance Measure.

## Data Availability

The data presented in this study are available on reasonable request from the corresponding author.

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
