# Peer review of "Exploring the Multidimensional Participation of Adults Living in the Community in the Chronic Phase following Acquired Brain Injury"

_ijerph, 2022, doi:10.3390/ijerph191811408_

Round 1

Reviewer 1 Report

The paper by Yosef det al deals with multidimensional participation of ABI patients in daily activities. The paper is interesting and has some strengths, as the use of ICF to categorize results, and the relationship between the perceived environmental accessibility and the different aspects of participation, as well as the use of COM .  

However some points remain difficult to understand. In the methods part there is no explanation BOUT THE Mayo-Portland Adaptability Inventory-4 despite the fact that it is deeply used for results.

It is not clear who conducted the participation measures and where (patient’s home? Outpatients clinics?)

In the methods authors stated they have used two scales for depression, probably due to the age gaps within the sample, but little information is given about how, why and if the two groups resulted significantly different by using different scales.

Another point that needs more clarification is the comparison between subjective and objective evaluation. How is possible to “explore the compatibility of the objective and subjective aspects of participation” if you analyzed occupational “subjective” problems by suing COPM, and then categorized results according to the 9 ICF domains, while you use a different classification for the objective evaluation?”.

As well, “Since the MPAI-4-P items and ICF domains do not exactly align; the occupational problems were reclassified by areas of participation according to the MPAI-4-P items.” (p.10)

As last methodological remark, from the statistics point of view, significance level are only given as p<0.05, but in this multi test setting, it should be advisable to provide actual significance level and in case also state if they are raw or corrected for multiple comparison results.

Reviewer 2 Report

Yosef at al., cross-sectionally investigate parameters of participation, as defined by ICF and subjective/objective indices, at the chronic state in a cohort of 25 acute brain injury patients with variable levels of disability. Naturally, this study reports differential self-reported participation domains in their study population and according to disability status. The results of this study is well supported bu their methodology and statistical approach; the rationale is well explained in the introduction and the discussion nicely synthesizes their findings in the context of the field.

Minor Comments:

Authors recommend that the findings of this study and others support the need in individualizing rehabilitative regimes according to objective patient disabilities, as well as taking into account their subjective disposition towards certain participation domains. If the authors could expand on real-world approaches, in their center or others, which utilize the information recorded in the study to personalize rehabilitation to improve upon standard practices; this would expand the relevance of their findings towards a broader audience.

Pre-clinical and clinical studies have reported that "loneliness" worsens near- and long-term outcomes in AIS patients, is it possible to extend the conclusions of these findings to elaborate on how objective/subjective participation overlap with social isolation in their patient group?

Overall the study is well done and supplements the current breadth of information towards assessing parameters of patients suffering the lasting consequences of acute brain injury to improve modalities of rehabilitation and recovery.

Round 2

Reviewer 1 Report

Authors satisfactorily replied to the some comments (e.g. the statistical issue) but their responses to some major issues are not in line with the concerns expressed in my review. I would ask a better discussion on these critical points. In particular point 4 and 5.
